# De-Escalation Strategies in HPV-Associated Oropharynx Cancer: A Historical Perspective with Future Direction

**DOI:** 10.3390/cancers16152733

**Published:** 2024-08-01

**Authors:** Clinton Wu, Paulina Kuzmin, Ricklie Julian

**Affiliations:** 1Department of Internal Medicine, University of Arizona, Tucson, AZ 85719, USA; clintonwu@arizona.edu (C.W.); pkuzmin@arizona.edu (P.K.); 2Department of Hematology and Oncology, University of Arizona, Tucson, AZ 85719, USA

**Keywords:** HPV-associated oropharyngeal cancer, de-escalation, ctDNA

## Abstract

**Simple Summary:**

Patients with locoregional head and neck cancer often receive radiation and/or surgery with curative intent. Chemotherapy is often used as a radiation sensitizer or adjunctively to improve treatment effect. However, these modalities are often associated with significant and potentially debilitating adverse events, which negatively impacts quality of life. HPV-associated oropharyngeal head and neck cancer has been shown to be more responsive to treatment compared to HPV-negative disease. Therefore, various de-escalation strategies have been under investigation in hopes of optimizing therapeutic response while also minimizing toxicity from treatment.

**Abstract:**

The incidence of HPV-related oropharyngeal cancers has increased in recent decades. While cure rates exceed those of HPV-negative head and neck cancers, both acute and long-term sequelae of chemotherapy, radiation and surgery have led to clinical investigation into de-escalation of treatment. De-escalation trials have sought to reduce long-term treatment-related morbidity by altering or omitting chemotherapy, reducing radiation, or incorporating less invasive surgical resection through transoral surgery. More recent approaches include the use of novel agents such as immunotherapy in place of cisplatin. With the advent of tumor-tissue-modified HPV DNA detection and monitoring in blood, new strategies incorporating this biomarker are being developed.

## 1. Introduction

Based on data from the American Cancer Society, there will be approximately 58,000 new cases of squamous cell carcinoma of the head and neck (SCCHN) diagnosed in 2024, with an estimated 12,000 deaths in the US alone [1]. Worldwide, there are more than 650,000 new cases of SCCHN annually with more than 350,000 deaths [2]. There is a predilection towards males, with an approximately 3-fold increase in the incidence of new cases compared to females. When analyzing the Surveillance Epidemiology, and End Results (SEER) registry, the incidence rates for males and females in 2021 were 17.7 and 6.7 per 100,000 respectively, with both genders exhibiting a comparable cumulative annual percentage increase of 0.6% since the early 2000s [3]. In 2021, non-Hispanic Native Americans had the highest incidence rate at 3.9 per 100,000, surpassing non-Hispanic white, Asian/Pacific Islander and Black patients. Hispanics had the lowest incidence rate overall. All groups showed increasing incidence rates per 100,000 apart from Non-Hispanic Black and Black patients. The median age at diagnosis for all genders is 64, although the incidence rate tends to be highest at >70 years of age. From data obtained between 2012 and 2021, approximately 53.7% of cases in males are initially diagnosed with stage III locoregional disease while in females, there is an approximately 40% split evenly between stage I/II localized and stage III locoregional disease [4]. 

The association with human papillomavirus (HPV) has shifted the paradigm on prognosis in head and neck cancer. Despite identifying more than 100 subtypes in the HPV family, there are approximately 12 high-risk HPV genotypes that have associated malignant potential: HPV-16, -18, -31, -33, -35, -39, -45, -51, -52, -56, -58, -59 and -68 [2]. However, the overwhelming majority of HPV-associated SCCHN is specifically from the HPV-16 genotype. 

Smoking, alcohol use and sexual behavior are the strongest modifiable risk factors most associated with the development of SCCHN [1,2]. Despite the recent decline in smoking and alcohol use in the US, the incidence rate has continued to rise owing to the increased prevalence of HPV-positive SCCHN. However, association with HPV portends a favorable prognosis and even alters staging in oropharynx cancers. Compared to traditional high-intensity options, new promising de-intensification strategies are now being actively studied to optimize treatment in HPV-positive oropharynx cancer.

Our article seeks to review the history and treatment of SCCHN as well as specifically focusing on current and ongoing advancements regarding de-escalation strategies in HPV-positive oropharyngeal squamous cell carcinoma (OPSCC). 

## 2. Biology

Part of the papillomaviridae family, HPV is classified as a non-enveloped double-stranded DNA virus [5]. HPV mainly infects either squamous and/or mucosal epithelium, particularly mucosal surfaces of the oropharyngeal and anogenital tract. HPV is known to cause a wide range of diseases including benign warts and papillomas, as well as genitourinary and head and neck cancers. The virus encodes two proteins, E1 and E2, that are responsible for DNA replication and it relies solely on cellular host machinery for the additional resources needed [6]. However, terminally differentiated keratinocytes lack the replicative machinery to advance from the G1 to the S phase, which would impact the virus’s ability to survive and maintain its proliferative state [7]. In order to circumvent this issue, HPV infects the basal layer of the epithelium and prevents terminal differentiation of keratinocytes through viral oncogenes E6 and E7 (Figure 1). The mechanism by which they maintain proliferative capacity is mainly studied in HPV-16 and HPV-18, two “high-risk” viruses with association for malignant potential. E6 was found to specifically degrade p53, a tumor suppressor gene, via a ubiquitin-dependent protease process [7,8]. Conversely, E7 binds to the unphosphorylated region of retinoblastoma (Rb), which subsequently releases the transcription factor E2F; thus abrogating the tumor suppressive effects and allowing for progression of the G1 to S phase of the cell cycle [5,6]. p16, also known as major tumor suppressor 1, binds cdk 4/6 and prevents association with cyclin D, preventing phosphorylation of Rb [9]. Under normal conditions, p16 functions to inhibit cell cycle progression, however, in the presence of HPV-associated E7 oncoproteins, p16 is unable to function and this inhibition is lost [10]. Therefore, p16 expression has been used as an important surrogate for HPV-associated cancers, although this is not a definitive correlation since multiple molecular pathways regulate p16 expression levels. Conversely, patients with head and neck cancer who have a history of tobacco and alcohol use have tumors that exhibit downregulation of p16 expression and this usually predicts a poor outcome [2]. 

## 3. History of SCCHN 

Dating back to the 1990s, the 5-year overall survival (OS) was relatively poor at approximately 54% [11]. Surgery was once considered the mainstay therapy as first described by Dr. George Crile in 1906 [12]. However, given the significant mortality and functional impairment of definitive surgery, many different surgical and chemoradiation approaches have been developed [13]. 

Radiation therapy (RT) was first introduced as a viable oncologic treatment in the late 19th century. In order to mitigate the significant post-surgical complications, the Radiation Therapy Oncology Group (RTOG) conducted the RTOG 73-03 trial, which attempted to evaluate the viability of combined surgery and radiation therapy on survival in stage II-IV disease [14]. Postoperative RT (60 Gy treated over ~6 weeks) was found to be superior compared to preoperative RT (50 Gy treated over ~5 weeks) in locoregional control (65% vs. 48%, *p* = 0.04) although there was no significant improvement in OS (38% vs. 33%, *p* = 0.10). However, this trial did find that the combined approach greatly improved locoregional control compared with patients who did not complete the study as indicated. As with surgery, post-radiation complications were also significant, ranging from dysphagia to edema with the potential for respiratory obstruction and tissue necrosis. 

To mitigate this, a new school of thought began to emerge utilizing a similar total dose of radiation but divided into smaller doses and treatment given at increased frequency. Termed hyperfractionation, various trials were conducted to address this issue. One of the first trials comparing hyperfractionation to standard fractionation was carried out by the European Organization for Research and Treatment of Cancer (EORTC). EORTC 22791 randomized 356 patients with oropharyngeal cancer ranging from T2 to T3N1 cancers to either arm [15]. The 5-year local control rate was significantly higher in the hyperfractionated arm (59% vs. 40%) and significance was achieved in T3, T3N0, and T3N1 cancers (*p* = 0.001, *p* = 0.03, and *p* = 0.01, respectively). The landmark trial was RTOG 90-03 in which 1113 patients were randomized to standard fractionation (70 Gy in 35 fractions over 7 weeks), hyperfractionation (82 Gy in 68 fractions over 7 weeks), accelerated fractionation (67 Gy in 42 fractions over 6 weeks including a 2-week rest), or accelerated fractionation with concomitant boost (72 Gy in 42 fractions over 6 weeks with 1.8 Gy per fraction per day given during the last 12 treatment days) [16]. Hyperfractionation and accelerated fractionation with boost resulted in improved locoregional control although there was no significant difference in disease-free survival (DFS) and OS. More importantly, the hyperfractionated and accelerated regimens were associated with increased acute events (defined by adverse effects <90 days from treatment start date) (*p* < 0.0001) predominately affecting cutaneous surfaces, the salivary gland, and the pharynx. Additionally, RTOG 9512 showed no statistical significance, but modest improvement in 5-year local control with hyperfractionation (78% vs. 70%, *p* = 0.14) even in T2 vocal cord carcinoma [17]. Interestingly, the EORTC 22791 trial also showed no significant 5-year local control in T2 disease, which implies that hyperfractionation may be more pronounced in more advanced stage II or stage III disease. Regardless, one of the most relevant adverse effects associated with these radiation treatments is salivary gland dysfunction, often leading to significant xerostomia, which can affect speech and swallowing, and increases the risk of oral infections. Since the early 2000s, an advanced radiation technique known as intensity-modulated radiotherapy (IMRT) attempted to circumvent collateral radiation to selective structures while maintaining potent efficacy. RTOG 00-22 was a phase II study in oropharyngeal cancer that initially showed significant locoregional control with a 2-year failure rate of 9% [18]. Of note, the seven patients exhibiting progression all had a history of smoking. The phase III randomized controlled PARSPORT trial formally compared conventional RT to parotid-sparing IMRT in developing grade 2 or above xerostomia within 12 months of treatment start date [19]. They found that there was a significant decrease in the development of grade 2 or above xerostomia in patients treated with IMRT (38% vs. 74%) without significant compromise in the efficacy of treatment. More recently, Nutting et al. used a modified version of IMRT, labeling it dysphagia-optimized (DO-IMRT), where the total radiation dose to the pharyngeal site was limited to 55 Gy, which, when compared to a total dose of 65 Gy to the primary site overall, showed improvement to patients swallowing function 12 months after RT [20]. These trials highlighted the utility of IMRT in maintaining treatment efficacy while also potentially preserving the patient’s quality of life. 

In contrast, chemotherapy did not initially blossom as a viable regimen for head and neck cancer until the introduction of cisplatin. In the early 1980s, multiple trials utilized induction therapy with cisplatin monotherapy or in combination with other anti-neoplastic agents such as bleomycin, vincristine, or 5-FU demonstrated very encouraging response rates overall [21,22,23]. 

In a phase II study, RTOG published one of the first trials using upfront chemoradiation in stage III and IV biopsy-proven SCCHN [24]. Of note, stage II SCC nasopharynx, base of tongue, and maxillary sinus were also included in this study. Cisplatin was administered at a dose of 100 mg/m^2^ on days 1, 22, and 43 of radiation therapy. ORR was very encouraging at approximately 70%, which was more pronounced as 82% of patients had characteristic stage IV disease at baseline. This ultimately led to a larger phase III Intergroup 0099 study where definitive chemoradiation (cisplatin 100 mg/m^2^ on days 1, 22, 43 with total dose RT of 70 Gy in 35 fractions over 7 weeks) with postradiotherapy cisplatin at 80 mg/m^2^ with 5-FU 1000 mg/m^2^ on days 1–4 administered every 4 weeks for a total of three cycles was studied against standard of care RT alone [25]. The 3-year progression-free survival (PFS) was 69% in the chemoradiation arm versus 24% in the RT-alone arm (*p* < 0.001). Secondary analysis using 3-year OS rates was also equally as significant (76% versus 46%, *p* < 0.001). More importantly, no reported patients had life-threatening toxicity from either arm, although patients in the chemoradiation arm developed more grade 3 leukopenia and gastrointestinal toxicity. However, one thing to mention is that 21 of 78 patients (27%) did not complete the chemoradiation protocol. Of those, 13 patients (17%) discontinued due to severe toxicity from treatment. This is in contrast with the RT arm, where 6 of 69 patients (8%) did not complete the protocol for reasons other than toxicity. Nevertheless, the significant survival advantage posed by the combined approach has fully entrenched this regimen as the standard of care in SCCHN. 

The crux of cytotoxic chemotherapy, particularly cisplatin, has been the potential for intolerable toxicity. Cisplatin has been commonly implicated in the development of neurotoxicity, nephrotoxicity, ototoxicity, and gastrointestinal-related issues, which can be life-threatening and more often than not require dose reductions and even permanent discontinuation before completing the recommended treatment course. Therefore, finding alternative strategies to revolutionize cancer therapy while also minimizing the toxicity profile remains the most crucial goal for all researchers and oncologists alike. 

## 4. De-Escalation Strategy

The association between HPV and head and neck cancer was first described by Syrjanen et al. in the 1980s when evaluating laryngeal cancer specimens in male patients [26]. However, HPV was only formally evaluated as an independent prognostic factor in the early 2000s. In a population-based study by Schwartz et al., they identified 40 patients with HPV-16-associated oropharyngeal cancer and found that there was a significant reduction in all-cause mortality (Hazard Ratio (HR) = 0.34, 95% CI 0.14 to 0.83) and disease-specific mortality (HR = 0.17, 95% CI 0.04 to 0.76), particularly when these patients received radiation therapy (*p* < 0.001) [27]. This subsequently led to one of the first formal prospective trials carried out by the Eastern Cooperative Oncology Group (ECOG) in 2008. ECOG 2399 was a phase II study comparing HPV-positive (HPV-16, -33, or -35) and HPV-negative tumors using two cycles of induction chemotherapy with carboplatin and paclitaxel, followed by 7 weeks of paclitaxel with RT [28]. In this study population, 95% of the HPV-positive tumors were also p16+. Response rates were assessed after induction and chemoradiation treatment. In general, there was a significant difference in the response rates between the two groups (82% versus 55%, *p* = 0.01; 84% versus 57%, *p* = 0.07 respectively). Survival outcomes also favored the HPV-positive tumors at the 2-year interval suggesting a favorable survival advantage in SCCHN with HPV-positive status. This finding was essentially corroborated in the RTOG 0129 study, which initially showed no benefit from accelerated versus standard fractionation regimen in combination with cisplatin. However, this trial also performed a retrospective analysis of HPV status which showed that HPV-positive cancer had significantly improved 3-year PFS and OS compared to HPV-negative cancer (OS 82.4% versus 57.1%, PFS 73.7% versus 43.4%) [29]. 

Prior to 2017, the traditional staging of oropharyngeal cancer by the American Joint Committee on Cancer (AJCC) implied that cancers staged up to IVB can be defined as locally advanced disease without evidence of metastasis. However, based on these aforementioned studies, locally advanced HPV-positive head and neck cancers showed improved prognosis and superior clinical benefit with high-dose chemoradiation. The significant discrepancy in the two subtypes has led AJCC to re-classify the 8th edition into HPV-positive and HPV-negative oropharyngeal cancer via surrogate p16 expression levels. Overall, p16-negative oropharyngeal cancers are similarly classified as the previous editions with the exception that the presence of extranodal extension automatically qualifies as N3, correlating with at least stage IVB disease. On the contrary, locally advanced HPV-positive oropharyngeal cancers are now formally recognized as stage III disease, while stage IV is exclusively reserved for metastatic spread. This change is reflected in the reassuring outcomes with HPV-positive oropharyngeal cancers. Treatment, however, has remained relatively unchanged despite downstaging and the toxicity profile remains a major barrier in treatment. Therefore, de-escalation strategies have been proposed to hopefully optimize patient care in locally advanced HPV-positive head and neck cancer (Figure 2). 

### 4.1. Decreasing Total Radiation Dose

Machtay et al. performed a comprehensive evaluation of three large RTOG trials (RTOG 91-11, 97-03, 99-14) that used chemoradiation as definitive therapy [31]. The purpose of this study was to assess the percentage of those patients who developed severe late toxicity, which was defined as either grade III/IV pharyngeal or laryngeal dysfunction, feeding tube dependence at least 2 years after initiation of the trial, or potential treatment-related death within 3 years. All three trial arms used RT at a dose of 70–72 Gy over 6–7 weeks with the exception of RTOG 97-03, which administered RT every other week. However, the chemotherapy regimen differed slightly in nature, with RTOG 91-11 administering three cycles of cisplatin 100 mg/m^2^ on weeks 1, 4, and 7 while RTOG 99-14 only administered two cycles on weeks 1 and 4. RTOG 97-03 had three separate arms: Arm 1, using daily cisplatin and 5-FU during the last 2 weeks of RT, Arm 2, using 5-FU and hydroxyurea, and Arm 3, consisting of once weekly of cisplatin and paclitaxel. They found that 99 of the 230 patients that met the criteria for this study had severe late toxicity, further reinforcing the notion that high-intensity concurrent chemoradiation carries as much of a significant risk as it does benefit. 

One of the first trials aimed at de-intensifying total radiation dose was conducted by Chera et al. [30]. In this phase II study, eligible patients were required to be HPV+ or p16+ with ≤10 pack-year smoking history, although a 10–30 pack per year history was permissible if they were abstinent for more than 5 years. These patients received RT at 60 Gy in 30 fractions over the course of 6 weeks with weekly cisplatin at 30 mg/m^2^. Primary analysis consisted of comparing pathologic CR (pCR) with a 3-year locoregional control rate of 86.4% demonstrated in the RTOG 0129 trial. The overall pCR was similar at 86% (98% at the primary site and 84% at the neck) with significantly fewer grade III and IV toxicities observed. This ultimately led to the landmark phase II study (NRG HN002) in which 306 p16+ patients with ≤10 pack per year smoking history were randomized to receive weekly cisplatin with IMRT (60 Gy in 30 fractions over 6 weeks) or IMRT alone (60 Gy in 30 fractions over 5 weeks) [32]. The purpose of this study was to formally evaluate whether both de-intensified regimens can be reasonable alternatives to conventional RT dosage. Based on data from the RTOG 0522 study, the primary objective was to show 2-year PFS > 85% with a target of 91% or greater and a post-1-year MD Anderson Dysphagia Inventory (MDADI) ≥ 60. The cisplatin + IMRT arm was found to be significant with a 2-year PFS of 90.5% (*p* = 0.04) and MDADI of 85.3. In the IMRT monotherapy arm, the 2-year PFS was not significant (87.6%, *p* = 0.23) although MDADI met the criteria. This ultimately supported the use of de-intensified radiation as a potential regimen to be further explored in a phase 3 trial. 

### 4.2. Cetuximab vs. Cisplatin

With the advent of molecular profiling techniques, driver mutations have been identified as potential contributing culprits for cancer development or relapse. Epidermal growth factor receptor (EGFR) was identified in the late 1990s as one potential driver that is commonly mutated in SCCHN [33]. EGFR is a transmembrane glycoprotein receptor that binds certain ligands such as EGF and tumor growth factor-alpha (TGF-α) and will eventually lead to cellular proliferation via various pathways [34]. Thus, Bonner et al. formally evaluated this with a phase III study comparing cetuximab (monoclonal antibody to EGFR) with RT versus RT alone in stage III/IV disease [35]. OS was 49 months in the cetuximab arm versus 29.3 months in the RT arm (*p* = 0.03). The most notable adverse events were grade III acneiform rash and infusion reactions. However, one question that was left unanswered was whether cetuximab would be more effective than cisplatin with RT. This was formally evaluated with the phase 3 RTOG 1016 trial which was a non-inferiority study comparing cetuximab (400 mg/m^2^ loading with 250 mg/m^2^ for 7 weekly doses) + RT (total 70 Gy) or cisplatin (100 mg/m^2^ for 2 doses) + RT (total 70 Gy) [36]. Overall, they found that the cetuximab arm was associated with significantly worsening OS, PFS, and locoregional failure compared to the cisplatin arm, although the prevalence of grade 3-4 toxicity was relatively similar in both groups. This finding was overall corroborated by the similar open-label phase 3 De-ESCALaTE HPV trial, which showed that while toxicity profiles were similar, patients on the cetuximab arm were associated with worsening 2-year OS [37]. This was further supported by the ARTSCAN III trial, in which the trial was prematurely discontinued after an interim analysis showed that cisplatin was overall superior to cetuximab (3-year OS was 88% versus 78%, *p* = 0.086; 3-year locoregional failure 23% versus 9%, *p* = 0.0036) [38]. Regardless, cetuximab remains a viable combination target with RT, particularly in cisplatin-ineligible patients. 

### 4.3. Induction Chemotherapy with Reduced-Dose Chemoradiation

RTOG 91-11 was one of the first studies utilizing induction chemotherapy in laryngeal cancer [39]. In this study, induction was accomplished with cisplatin and 5-FU prior to definitive radiation. The primary outcome, laryngeal preservation, was ultimately inferior to the chemoradiation arm and was not significant when compared to RT alone. However, HPV status was unknown at the time. The first trial to demonstrate the correlation between HPV-positive status and PFS with induction chemotherapy was shown in the E2399 trial [40]. This ultimately led to the E1308 trial, a phase II study aimed and determining whether reduced-dose radiation can lower toxicity while maintaining comparable PFS [41]. This trial was designed to administer three cycles of induction cisplatin, paclitaxel, and cetuximab followed by either high-dose or reduced-dose (69 Gy vs. 54 Gy) radiation with weekly cetuximab depending on response with induction therapy. Patients who had a complete clinical response at the primary site as determined by radiographic imaging carried out within 14 days post-induction therapy were assigned to the reduced-dose radiation arm. Overall, the 2-year PFS (80%) and OS (94%) were relatively encouraging with the reduced-dose arm. Moreover, patients on the reduced-dose arm had fewer grade 3 toxicities including mucositis and dysphagia. This ultimately led to the Quarterback trial, which aimed to use reduced-dose chemoradiation in patients who responded to induction chemotherapy [42]. Overall, the 3-year PFS and OS was >80% in both groups and overall comparable, although comparing between standard and reduced dose was difficult to interpret as it was a small study with only 22 eligible patients. A follow-up of three sequential studies as part of the Quarterback trials was recently presented at the American Society of Clinical Oncology (ASCO) in 2023, which showed that the reduced-dose chemoradiation regimen maintained potency with improved quality of life [43].

### 4.4. Immunotherapy

Immune checkpoint inhibitors have altered the landscape of cancer therapy given the potential to reconstitute immune-mediated effector function against cancer cells. Antibodies disrupting the programmed cell death-1/programmed cell death ligand-1 axis (PD-1/PD-L1) have been the foundational backbone of immunotherapy treatments overall. Immunotherapy has shown activity in HNSCC, including p16+ disease. KEYNOTE-012 was a small phase 1b study that studied the effects of pembrolizumab in metastatic or recurrent head and neck cancer [44]. Of the 56 patients analyzed, 20 were HPV-positive and 36 were HPV-negative. Based on an investigator review, the ORR for the study population was 21%. When subcategorizing between HPV-positive and HPV-negative status, there was a trend towards improved response with HPV-positive status (ORR 25% vs. 19%) although no formal statistical comparison was performed. One of the first trials using immunotherapy in locally advanced SCCHN was the JAVELIN Head and Neck 100 trial, which assessed whether the addition of avelumab with chemoradiation could improve survival advantage compared with chemoradiation alone [45]. Unfortunately, the trial was formally discontinued at the interim analysis as there was no improvement in PFS, OS, or ORR when comparing the two groups. In a subgroup analysis, HPV status did not influence the outcome and surprisingly, even trended towards favoring the placebo arm (HPV-positive: HR 1.26 (0.74–2.15), HPV-negative: HR 1.16 (0.86–1.57). However, one noted advantage was that the addition of avelumab did not increase the incidence of adverse events (mucositis, anemia, leukopenia, dysphagia, and serious AEs) overall. As mentioned previously, cisplatin with radiation has been one of the most effective strategies in the treatment of locally advanced head and neck cancer. Sadly, the toxicity profile of cisplatin remains a challenge and patients are often unable to initiate or even complete the recommended cisplatin regimen leading to suboptimal efficacy. Therefore, the NRG-HN004 trial compared the use of cetuximab + RT to durvalumab + RT in patients with locally advanced head and neck cancer who were cisplatin-ineligible. Patients with p16-positive disease were included [46]. The goal was to determine whether immunotherapy would be more effective than cetuximab in prolonging PFS. Regrettably, durvalumab did not improve either 2-year PFS or OS (51% vs. 66%, HR 1.47 (0.86–2.52), *p* = 0.92) (70% vs. 78%, HR 1.21 (0.63–2.31), *p* = 0.72) and was even correlated with worsening locoregional failure compared to cetuximab (32% vs. 16%, HR 2.17 (1.00–4.69), *p* = 0.04). Given these results, the trial ultimately did not proceed to a phase III study. Although immunotherapy has been found to be inferior compared to platinum chemotherapy and cetuximab, it has shown some efficacy in the treatment of head and neck cancers and remains under investigation as a de-escalation strategy. The NRG-HN005 trial is an active study currently seeking to determine whether the addition of immunotherapy to low-dose RT could maintain therapeutic benefit with less toxicity profile in low-risk p16-positive oropharynx cancers. 

### 4.5. Circulating Tumor DNA 

Serologic biomarkers remain an important tool to assist in diagnosis and prognostic implications, as well as evaluating treatment effect. More recently, measuring circulating tumor DNA (ctDNA) has offered clinicians an additional minimally invasive modality to monitor the efficacy of treatment. Specifically, in the setting of HPV-associated oropharyngeal cancer, there has been interest in using circulating tumor HPV DNA (ctHPVDNA) as a biomarker for HPV-associated oropharyngeal squamous cell carcinoma for post-treatment surveillance. Cao et al. were the first to demonstrate that it was possible to detect circulating HPV DNA in a majority of patients using quantitative polymerase chain reaction (qPCR) [47]. Conversely, ctHPVDNA was not detected in HPV-negative cancers. In a subset of 14 patients with HPV-positive disease, they were able to serially demonstrate undetectable levels of plasma HPV DNA after receiving chemoradiation. Furthermore, four patients eventually relapsed, and this also correlated with a subsequent rise in HPV DNA levels. Chera et al. evaluated 115 patients with nonmetastatic HPV-associated OPSCC treated with CRT and monitored with blood samples for ctHPVDNA. In this cohort, 28 individuals developed a positive ctHPVDNA during post-treatment surveillance, with 15 confirmed biopsy recurrences. They reported a negative predictive value (NPV) of 100% and a positive predictive value (PPV) of 94% when using two consecutive positive ctHPVDNA tests [48]. Berger et al. conducted the first prospectively designed retrospective consecutive clinical case series (*N* = 1076) in evaluating circulating tumor-tissue-modified HPV DNA with an NPV finding of 95%. Their major conclusion was the potential for TTMV-HPV DNA testing as a surveillance tool, as the blood test often yielded positive results before routine clinical or imaging exams [49]. Given the sensitivity of ctHPVDNA, Chera and colleagues postulate whether a favorable ctHPV16DNA profile could lead to treatment de-escalation. Monitoring ctHPVDNA, at the very least, may help personalize treatment in efforts to minimize treatment-associated morbidity. Currently, the ReACT study is a phase 2 clinical trial actively enrolling and investigating de-escalation treatment with the use of a blood sample measuring ctDNA levels along with favorable clinical characteristics [50]. The study is aimed to take place over 5 years with 75 individuals enrolled. Additionally, there is a phase 2 study (NCT05307939) currently in active recruitment seeking to determine whether patients who undergo definitive surgery can minimize their need for adjuvant RT by monitoring ctHPVDNA levels. Lastly, a large prospective study (NCT05541016) is attempting to assess the effectiveness of utilizing ctHPVDNA in four different cohorts: favorable, unfavorable, high risk, and chemoradiation. The primary objective is to determine whether utilizing ctHPVDNA can help select the appropriate treatment intensity regimen for each cohort. The monitoring of ctHPVDNA through blood samples or potentially saliva sampling [51] emerges as a crucial biomarker, not only aiding in surveillance but also potentially shaping de-escalation treatment approaches. 

### 4.6. HPV16 Subgroups

HPV16 subgroups, or phylogenetic variants, have significant implications for head and neck squamous cell carcinomas. HPV16 is classified into four main phylogenetic lineages (A–D), with each lineage exhibiting distinct biological and clinical behaviors [52]. Lineage A, often referred to as the “European variant,” is predominantly found in HNSCCs. Studies have shown that specific inter-typic variants within this lineage, such as those with the T350G polymorphism in the E6 gene, can influence clinical outcomes. For instance, the T350G variant has been associated with differences in disease-free survival and B-cell-mediated immune responses [53]. In contrast, lineages B, C, and D, previously termed “non-European variants,” are more commonly associated with high-grade cervical lesions and cancers. Zhang et al. identified two distinct subtypes of HPV-positive HNSCCs: HPV-KRT and HPV-IMU. These subtypes differ in gene expression, viral integration patterns, and genomic alterations. HPV-KRT is characterized by keratinocyte differentiation and higher PIK3CA mutation rates, while HPV-IMU shows strong immune response and mesenchymal differentiation [54]. However, their role in HNSCCs is less well-characterized, necessitating further large-scale studies to elucidate their impact on disease development and outcomes [55].

### 4.7. Intratumor Heterogeneity

Intratumor heterogeneity (ITH) can manifest as variations in DNA mutations, somatic copy number alterations (SCNAs), and gene expression profiles across different regions of the tumor. Studies have shown that ITH can significantly impact tumor behavior, treatment response, and prognosis. For instance, Götte et al. demonstrated that intratumoral genomic heterogeneity is prevalent in HNSCC, with notable differences in chromosomal ploidy and genetic alterations between primary tumors and corresponding metastases [56]. Similarly, Ledgerwood et al. found that the degree of mutational heterogeneity varies by primary tumor subsite, with laryngeal and floor-of-mouth tumors exhibiting higher unique single nucleotide variants (SNVs) compared to oral tongue tumors [54]. Gram et al. conducted a prospective study that included 33 whole tumor specimens from 28 patients with primary or recurrent HNSCC and conducted a genetic analysis of somatic copy number alterations (SCNAs), focusing on 45 preselected genes of interest. They found distinct variations with recurrent tumors exhibiting higher levels of variation in SCNAs than primary tumors [57]. Moreover, Mroz et al. reported that higher levels of genetic heterogeneity, as measured by the mutant-allele tumor heterogeneity (MATH) score, are associated with decreased overall survival in HNSCC patients [58].

### 4.8. NF-kB Activity and Mutations in NF-kB Regulators

There have been additional investigations into nuclear factor-kappa B (NF-κB) and its role in the pathogenesis of HPV-related diseases [59]. NF-κB is a family of transcription factors involved in immune responses, inflammation, and cancer progression. In HPV-associated HNSCC, NF-κB overactivation is linked to better clinical outcomes. Tumors with defects in TRAF3 or CYLD, which are negative regulators of NF-κB, show increased NF-κB activity and improved survival rates. This suggests that NF-κB activity may serve as a prognostic marker and potential therapeutic target in these cancers [60]. Additionally, HPV E6 protein can inactivate CYLD, leading to prolonged NF-κB activation under hypoxic conditions, which is associated with adverse clinical outcomes [61]. Additional research may be useful in stratifying by mutations in Nf-kB.

## 5. Conclusions

The standard by which oncologists approach the treatment of cancer has shifted significantly. Historically, the diagnosis of cancer and associated poor prognosis has led to aggressive treatment regimens. However, treatment toxicity has been questioned in cancers with high survival rates. The identification of HPV as a favorable prognostic indicator in oropharyngeal cancer has created a new state of equipoise in clinical research. As the paradigm for cancer management has shifted to optimizing care, many de-escalation strategies have gained attention in HPV-associated oropharyngeal cancer. This review article highlights different de-escalation strategies hoping to maximize treatment efficacy while also limiting toxicity in an attempt to maintain quality of life. 

## Figures and Tables

**Figure 1 cancers-16-02733-f001:**
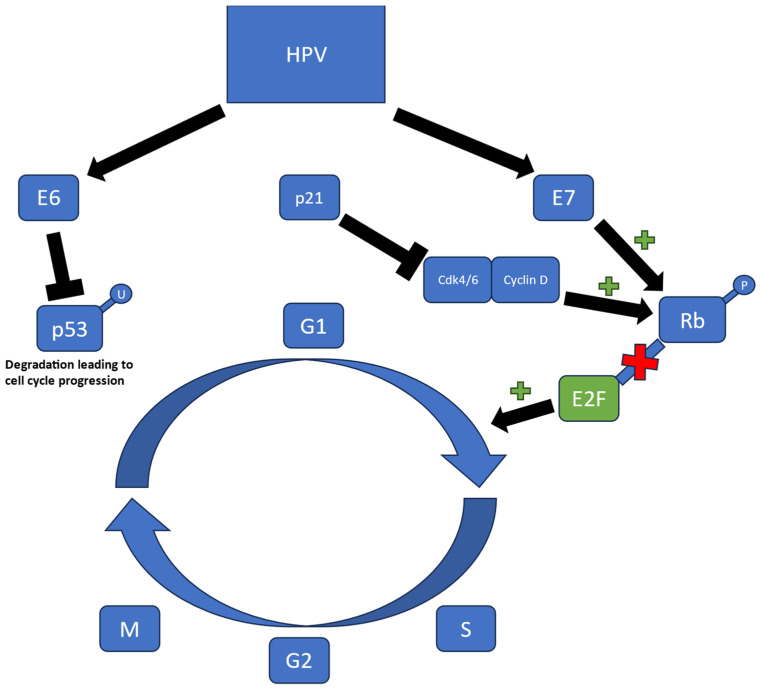
How HPV+HNSCC regulates the cell cycle.

**Figure 2 cancers-16-02733-f002:**
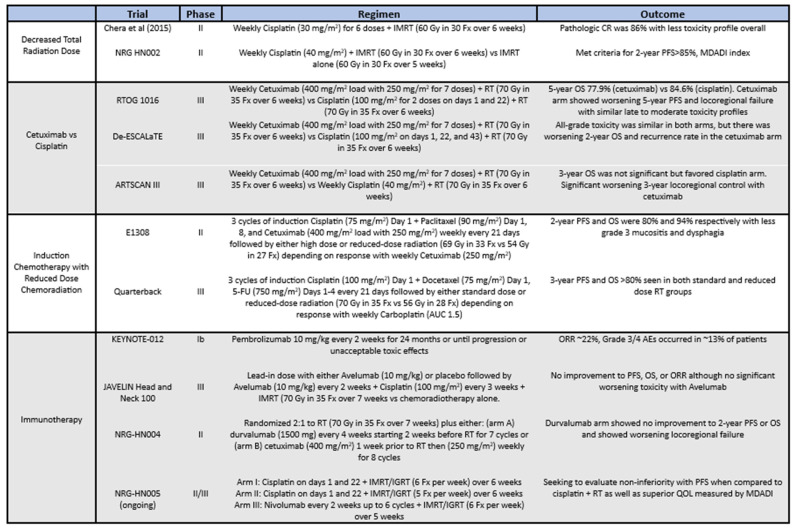
List of the various trials (previous and ongoing) aimed at de-escalation strategies for HPV p16+ oropharyngeal cancer [30].

## Data Availability

The data presented in this study are available on request from the corresponding author.

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
