# Peer review of "De-Escalation Strategies in HPV-Associated Oropharynx Cancer: A Historical Perspective with Future Direction"

_cancers, 2024, doi:10.3390/cancers16152733_

Round 1

Reviewer 1 Report

Comments and Suggestions for Authors

The authors present a manuscript that outlines de-escalation in treatment of HPV+ oropharyngeal cancer (OPC) that focuses heavily on efforts to decrease radiation or change radiation sensitizers. The authors also mention the potential for circulating tumor DNA for de-escalation efforts.

Although well written, the manuscript shortchanges de-escalation strategies that rely on induction/neoadjuvant and surgery as methods to minimize radiation (e.g. ECOG/ACRIN 3311 and 1308 trials). In addition to circulating HPV DNA, potential biomarkers that may be useful to guide de-escalation for this disease including HPV genome integration, HPV16 subtypes, intratumor heterogeneity, NF-kB activity, mutations in NF-kB regulators, and estrogen receptor expression should be reviewed.

The manuscript also lacks a conclusion paragraph that would be useful to summarize major points of the manuscript.

Other points that should be addressed:

·         Line 43. HPV is known to cause a wide range or etiologies ….

HPV is the etiology of illnesses or diagnoses

·         Lines 59-60 are confusing/inaccurate. p16 normal function is to inhibit cell cycle progression, but in the presence of HPV E7 oncoprotein, p16 cannot perform this function

·         Lines 65-66 are confusing/inaccurate. Mutational burden does not alter p16 expression and p16 loss primary prognostic impact is defining HPV-associated vs. smoking associated HNSCC

·         Line 105. Trial also showed no significant 5-year local control in T2 disease

a.      Trial also showed no significant improvement in 5-year local control in T2 disease

·         In the section “Decreasing Total Radiation Dose” when discussing the regimen with radiation of 60Gy with weekly cisplatin 30mg/m2 (as in HN002), the authors should incorporate the latest news related to HN005 where this arm was halted for lack of efficacy and the other experimental arm is being advanced to phase 3.

·         For discussion of trial results throughout, the authors should clarify if the trials were inclusive of both HPV-associated and unassociated disease or only included HPV-associated OPC.

Author Response

Comments: 

The authors present a manuscript that outlines de-escalation in treatment of HPV+ oropharyngeal cancer (OPC) that focuses heavily on efforts to decrease radiation or change radiation sensitizers. The authors also mention the potential for circulating tumor DNA for de-escalation efforts.

Although well written, the manuscript shortchanges de-escalation strategies that rely on induction/neoadjuvant and surgery as methods to minimize radiation (e.g. ECOG/ACRIN 3311 and 1308 trials). In addition to circulating HPV DNA, potential biomarkers that may be useful to guide de-escalation for this disease including HPV genome integration, HPV16 subtypes, intratumor heterogeneity, NF-kB activity, mutations in NF-kB regulators, and estrogen receptor expression should be reviewed.

The manuscript also lacks a conclusion paragraph that would be useful to summarize major points of the manuscript.

Other points that should be addressed:

  • Line 43. HPV is known to cause a wide range or etiologies ….

HPV is the etiology of illnesses or diagnoses

  • Lines 59-60 are confusing/inaccurate. p16 normal function is to inhibit cell cycle progression, but in the presence of HPV E7 oncoprotein, p16 cannot perform this function
  • Lines 65-66 are confusing/inaccurate. Mutational burden does not alter p16 expression and p16 loss primary prognostic impact is defining HPV-associated vs. smoking associated HNSCC
  • Line 105. Trial also showed no significant 5-year local control in T2 disease
  1. Trial also showed no significant improvement in5-year local control in T2 disease
  • In the section “Decreasing Total Radiation Dose” when discussing the regimen with radiation of 60Gy with weekly cisplatin 30mg/m2 (as in HN002), the authors should incorporate the latest news related to HN005 where this arm was halted for lack of efficacy and the other experimental arm is being advanced to phase 3.
  • For discussion of trial results throughout, the authors should clarify if the trials were inclusive of both HPV-associated and unassociated disease or only included HPV-associated OPC.

Response: 

Thank you for your comments. The ECOG/ACRIN 3311 and 1308 trials were added. We also added an additional section to address investigational biomarkers to guide de-escalation, specifically reviewing HPV genome integration, HPV16 subtypes, intratumor heterogeneity and NF-kB. 

Conclusion paragraph has been added. 

Line 43 corrected. 

Line 59-60 corrected. 

Line 65-66 clarified/corrected. 

Added HN005 information. 

Reviewer 2 Report

Comments and Suggestions for Authors

A question of de-escalation strategy to treat HPV-associated oropharynx was asked just after  discovery of  lower risk of disease progression as compared with non-HPV cancer usually connected with cigarette smoking and alcohol abusing. It looks quite reasonable to use saving therapy for HPV(-)  oral cancer.  Why the problem is still not solved? It is connected with patients getting a saving therapu but not cured sufficiently that opens a way for grievance and a field for lawyers.

The authors have written their text cautiously referring mostly to newest findings.  Therefore conclusions are also equally prudent. Anyway in my opinion the article deserves publication.

Minor remarks:

1.. I have never seen so short address of the authors.

2. Include reference number before full-stop

Author Response

A question of de-escalation strategy to treat HPV-associated oropharynx was asked just after  discovery of  lower risk of disease progression as compared with non-HPV cancer usually connected with cigarette smoking and alcohol abusing. It looks quite reasonable to use saving therapy for HPV(-)  oral cancer.  Why the problem is still not solved? It is connected with patients getting a saving therapu but not cured sufficiently that opens a way for grievance and a field for lawyers.

The authors have written their text cautiously referring mostly to newest findings.  Therefore conclusions are also equally prudent. Anyway in my opinion the article deserves publication.

Minor remarks:

1.. I have never seen so short address of the authors.

  1. Include reference number before full-stop

Response: De-escalation therapy has been used in other cancer types, such as hormone-positive breast cancer, where cure rates are exceptionally high and long-term morbidity and mortality due to therapy merit investigation of strategies that may maintain high cure rates but offset treatment adverse events. There have been many clinical trials to attempt to de-escalate therapy in HPV associated head and neck cancer, and these are outlined in this review. We will edit the reference number/full stop to meet formatting. 

Reviewer 3 Report

Comments and Suggestions for Authors

I have reviewed this interesting well-written review on de-escalation strategies in HPV-positive oropharyngeal cancer. The authors selected 51 articles and the main topics were covered. The text presented a good conducting line. 

Author Response

Comments: 

I have reviewed this interesting well-written review on de-escalation strategies in HPV-positive oropharyngeal cancer. The authors selected 51 articles and the main topics were covered. The text presented a good conducting line. 

Reply: 

Thank you for your comments. 

Reviewer 4 Report

Comments and Suggestions for Authors

The authors provide a brief overview of Squamous cell carcinoma of the head and neck (SCCHN) epidemiology in some populations, they describe molecular basis of its association with human papillomavirus (HPV), provide an info on the history of SCCHN studies including in relation to HPV, introduce concept of treatment de-escalation, discuss some of the current therapeutic approaches and describe some low-invasive biomarkers of SCCHN which can guide the curative strategies with treatment de-escalation.

Please find below my comments and suggestions.

Regarding the abstract: “tumor-tissue modified HPV DNA detection “ - what does it mean? Perhaps tumour tissue-derived HPV DNA detection?

Aforementioned, the authors start from providing a brief overview of SCCHN epidemiology data. However, their data are biased towards one particular population/country. To avoid such bias please provide an information on SCCHN epidemiology worldwide.

Line 15 In 2021, non-Hispanic Native Americans had the highest incidence rate, surpassing non-Hispanic whites while Hispanics had the lowest incidence rate overall” - what about information regarding other races? Why did the authors overlooked data regarding African-American, Asian, for example?

Furthermore, I suggest starting this review from a brief introduction of pathogenesis of SCCHN (cells of origin, genetics, clinical and molecular aspects, etc).

Line 56. “p16, also known as major tumor suppressor 1, is located on chromosome 9q21 and specifically binds cdk 4/6” - this sentence is not correct, the authors start talking about p16 protein and, next, change the subject and describe location of the gene that encodes p16, and next continue talking about protein. Obviously, p16 protein, the one which binds cdk4/6, is not located on chromosome 9q21, it's a gene localization.

Given the ability of HPV-associated E7 oncoproteins effect on Rb” - please rewrite this odd-sounding sentence.

Therefore, p16 expression has been used as an important surrogate for HPV-associated cancer” - perhaps a biomarker, not a surrogate.

Dating back to the 1990s, the 5-year overall survival (OS) was relatively poor at approximately 54%” - OS for SCCHN

Apart from circulating tumor-derived free DNA, what are the other biomarkers for non-invasive surveillance, the ones which can be used to establish novel de-escalation treatment strategies?

What information can you provide about the HPV-status-dependent and HPV-status-specific secretome/metabolome of the tumor, which can be used in a context of de-escalation treatment approaches?

What about tumor-associated microbiome/microbiota, which may vary depending on HPV status of the tumor?

A table concisely summarizing tumor-associated microbiome/microbiota, it's association with HPV-status, and corresponding curative approaches (including, if applicable, de-escalation approaches) would be nice addition to the text.

The conclusion sub-section is missing, please add it to the revised version of the review.

Author Response

Comments: 

The authors provide a brief overview of Squamous cell carcinoma of the head and neck (SCCHN) epidemiology in some populations, they describe molecular basis of its association with human papillomavirus (HPV), provide an info on the history of SCCHN studies including in relation to HPV, introduce concept of treatment de-escalation, discuss some of the current therapeutic approaches and describe some low-invasive biomarkers of SCCHN which can guide the curative strategies with treatment de-escalation.

Please find below my comments and suggestions.

Regarding the abstract: “tumor-tissue modified HPV DNA detection “ - what does it mean? Perhaps tumour tissue-derived HPV DNA detection?

Aforementioned, the authors start from providing a brief overview of SCCHN epidemiology data. However, their data are biased towards one particular population/country. To avoid such bias please provide an information on SCCHN epidemiology worldwide.

Line 15 In 2021, non-Hispanic Native Americans had the highest incidence rate, surpassing non-Hispanic whites while Hispanics had the lowest incidence rate overall” - what about information regarding other races? Why did the authors overlooked data regarding African-American, Asian, for example?

Furthermore, I suggest starting this review from a brief introduction of pathogenesis of SCCHN (cells of origin, genetics, clinical and molecular aspects, etc).

Line 56. “p16, also known as major tumor suppressor 1, is located on chromosome 9q21 and specifically binds cdk 4/6” - this sentence is not correct, the authors start talking about p16 protein and, next, change the subject and describe location of the gene that encodes p16, and next continue talking about protein. Obviously, p16 protein, the one which binds cdk4/6, is not located on chromosome 9q21, it's a gene localization.

Given the ability of HPV-associated E7 oncoproteins effect on Rb” - please rewrite this odd-sounding sentence.

Therefore, p16 expression has been used as an important surrogate for HPV-associated cancer” - perhaps a biomarker, not a surrogate.

Dating back to the 1990s, the 5-year overall survival (OS) was relatively poor at approximately 54%” - OS for SCCHN

Apart from circulating tumor-derived free DNA, what are the other biomarkers for non-invasive surveillance, the ones which can be used to establish novel de-escalation treatment strategies?

What information can you provide about the HPV-status-dependent and HPV-status-specific secretome/metabolome of the tumor, which can be used in a context of de-escalation treatment approaches?

What about tumor-associated microbiome/microbiota, which may vary depending on HPV status of the tumor?

A table concisely summarizing tumor-associated microbiome/microbiota, it's association with HPV-status, and corresponding curative approaches (including, if applicable, de-escalation approaches) would be nice addition to the text.

The conclusion sub-section is missing, please add it to the revised version of the review.

Response: “Tumor-tissue modified HPV DNA" is the terminology used by Naveris is their in HPV circulating DNA test, NavDx. It is the only commercially available test of this kind currently and while the name of the test is lengthy, it is meant to describe that it is not just detecting HPV DNA but rather HPV DNA that has been modified by a cancer cell. “Tumor-tissue modified HPV DNA" is the term that has been used across publications validating this test. 

We will add information on world-wide data and be inconclusive of other races and ethnicities. 

Line 56 will be corrected, thank you for pointing this out. 

The term "surrogate" is often used in the literature to describe how p16 expression is used in regard to HPV. 

An additional section was added to address biomarkers being investigated to aid in de-escalation. 

The conclusion has been added. 

Round 2

Reviewer 4 Report

Comments and Suggestions for Authors

It would be much appreciated if you could answer briefly to my previously raised questions (see below)

Apart from circulating tumor-derived free DNA, what are the other biomarkers for non-invasive surveillance, the ones which can be used to establish novel de-escalation treatment strategies?

What information can you provide about the HPV-status-dependent and HPV-status-specific secretome/metabolome of the tumor, which can be used in a context of de-escalation treatment approaches?

What about tumor-associated microbiome/microbiota, which may vary depending on HPV status of the tumor?

Additionally, please fix the typo in the added text:

Line 407. "NK-kB regulators"

Author Response

Comments 1: Apart from circulating tumor-derived free DNA, what are the other biomarkers for non-invasive surveillance, the ones which can be used to establish novel de-escalation treatment strategies?

Response 1: There have been no established, or guidelines-supported, biomarkers for de-escalation. Staging and risk stratification based on RTOG 0129 risk groups have been used as the basis for de-escalation and circulating HPV tumor DNA has newly been incorporated into study schemas. However, to date, none of these strategies have effectively impacted or changed guidelines. Further studies with circulating tumor DNA and exploratory analyses of predictive biomarkers are underway.

Comments 2: What information can you provide about the HPV-status-dependent and HPV-status-specific secretome/metabolome of the tumor, which can be used in a context of de-escalation treatment approaches?

Response 2: This has not been used as a basis for clinical de-escalation of therapy. 

Comments 3: What about tumor-associated microbiome/microbiota, which may vary depending on HPV status of the tumor?

Response 3: This has not been used as a basis for clinical de-escalation of therapy. For comments 2/3, we can include some of the data for prognosis if desired but that has not been used as a strategy for de-escalation. 

Round 3

Reviewer 4 Report

Comments and Suggestions for Authors

I suggest, you might want to include into discussion some info from your responses to questions 1,2,3, as some of the readers might have such questions in their mind while reading your manuscript. It's up to you of course.

The font in the Figure 2 is a bit too small and fuzzy, it will be impossible to read in a printout format (and many people still read printouts, to make a notes on the text). I recommend increasing font.